# CTRL123: CONSISTENT NOVEL VIEW SYNTHESIS VIA CLOSED-LOOP TRANSCRIPTION

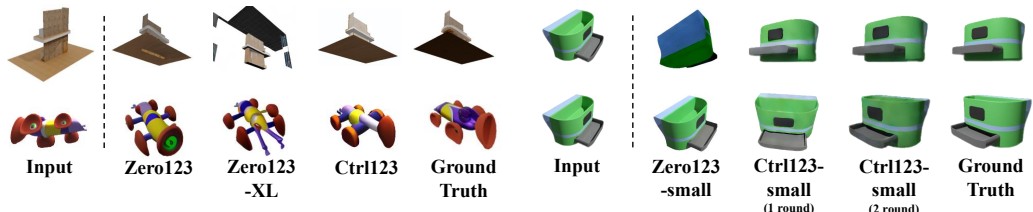

(a) On the **training dataset**, both Zero123 and Zero123-XL exhibit limited alignment capability and excessive diversity in their generated images.

(b) On a small dataset, Ctrl123-small over-fits better, which proves the stronger alignment capability of Ctrl123's training strategy (see details in Section 4.2).

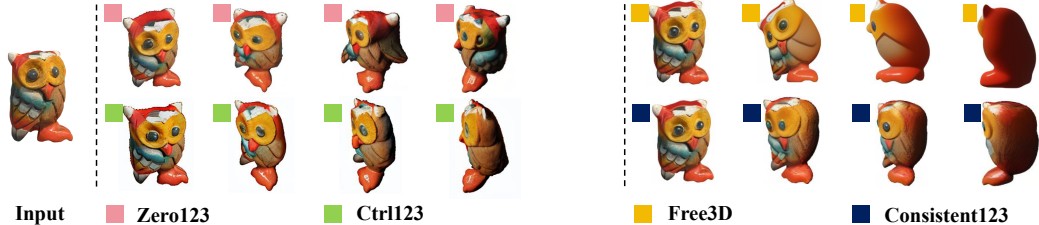

(c) Maintaining the original task settings, Ctrl123 generates more consistent multi-view images.

Figure 1: In our work, we aim to achieve multi-view consistency without changing the task settings and training dataset of Zero123 (Liu et al., 2023a). We root the multi-view inconsistency problem to the excessive diversity of diffusion models, which results from the weak alignment capability with ground truth images of Zero123's (Liu et al., 2023a) training strategy (as shown in Figure 1(a)), and solve it by adding the additional constraint on generated images with the help of **closed-loop** transcription. As shown in Figure 1(c), keeping the task setting of generating arbitrary novel views, Ctrl123 can generate better consistent multi-views than other baselines.

## ABSTRACT

Based on the success of large image diffusion models, multi-view diffusion models have demonstrated remarkable zero-shot capability in novel view synthesis (NVS). However, the pioneering work Zero123 (Liu et al., 2023a) struggles to maintain consistency across generated multiple views. While recent modifications in model and training design have improved multi-view consistency, they often introduce new limitations, such as restricted fixed view generation or reliance on additional conditions. These constraints hinder the broader application of multi-view diffusion models in downstream tasks like 3D reconstruction. We identify the root cause of inconsistency as the excessive diversity inherent in generative models utilized for the NVS task. To address this, we aim to utilize the stronger supervise information to better alignment with ground truth images to constrain the diversity, and propose Ctrl123, a **closed-loop** transcription-based multi-view diffusion method that enforces alignment in the CLIP (Radford et al., 2021) patch feature space. Extensive experiments demonstrate that Ctrl123 excels in **arbitrary** novel view generation, significantly improving multi-view consistency compared to existing methods.

# 1 INTRODUCTION

Recent advancements in novel view synthesis (NVS) have sparked considerable excitement on 3D generation (Poole et al., 2022; Wang et al., 2023c; Liu et al., 2024; Tang et al., 2024; Wu et al., 2024; Xu et al., 2024). The pioneering work of multi-view diffusion models, which achieve NVS, Zero123 (Liu et al., 2023a), utilizes 2D latent diffusion models to directly generate images of novel views. Although Zero123 (Liu et al., 2023a) demonstrates impressive zero-shot and open-world capabilities, it still encounters a multi-view inconsistency problem among the generated views, particularly on wild images, as shown in Figure 1(c). Subsequent studies (Shi et al., 2023a; Liu et al., 2023b; Shi et al., 2023b; Long et al., 2023; Zheng & Vedaldi, 2024; Voleti et al., 2024; Yang et al., 2024) attempt to address this inconsistency problem, but alter the original task settings and introduce new limitations. Zero123++(Shi et al., 2023a) and SyncDreamer(Liu et al., 2023b) generate images of multiple views simultaneously, but they are restricted to generating images of fixed views. TOSS (Shi et al., 2023b) and Free3D (Zheng & Vedaldi, 2024) mitigate the inconsistency issue by leveraging additional conditions (text descriptions and absolute camera extrinsic, respectively), but the need of these additional conditions constrains the application scenarios of multi-view diffusion models. SV3D (Voleti et al., 2024) and Hi3D (Yang et al., 2024) seek to address the inconsistency problem using a video generation model, yet SV3D demonstrates poor performance when dealing with large relative elevation transformations, as illustrated in Figure 6, and Hi3D requires additional information about the camera's absolute elevation and lacks the capability to generate novel views with relative elevation transformations.

Although the modified task settings, like generating fixed-view multiple views, can enhance consistency, the original task setting of Zero123 (Liu et al., 2023a) enables multi-view diffusion models to generate arbitrary views, allowing them to be used in all downstream reconstruction models. Current reconstruction models (Liu et al., 2024; Tang et al., 2024; Wu et al., 2024; Xu et al., 2024) use multiple views generated by multi-view diffusion models as input, and different reconstruction models employ different multi-view settings: while InstantMesh (Xu et al., 2024) inputs fixed multi-views generated by Zero123++ (Shi et al., 2023a), Unique3D (Wu et al., 2024) employs four orthographic views as input. Therefore, unlike most recent works (Shi et al., 2023a; Liu et al., 2023b; Zheng & Vedaldi, 2024; Voleti et al., 2024), our approach focuses on addressing the inconsistency problem without altering the original task settings. We hypothesize that the inconsistency problem stems from the excessive diversity of generated images. While this diversity is beneficial for text-to-image generation where only one single image is desired, it undermines the multi-view consistency which is crucial for NVS task. The excessive diversity among generated multiple views results in inconsistency, leading to poor 3D reconstruction performance.

Learning to generate images identical to ground truth (GT) images is a simple yet effective objective for controlling the excessive diversity. This objective does not encourage diversity, and we refer to the learning process as fitting GT images. However, recent multi-view diffusion models typically fine-tune pre-trained diffusion models with the original score matching (SM) training strategy, which is designed to learn a distribution with inherent diversity, instead of fitting GT images. In fact, the traditional score matching (SM) training strategy has limited ability to fit GT images effectively. To support this claim, we train a new model, denoted as Zero123-small, with the same training strategy and model configuration as Zero123 (Liu et al., 2023a) on a small dataset containing only 25 objects (see Section 4.2, Figure 1(b) and Table 1 for more details). Over-fitting on such a small dataset should be an easy task, yet Zero123-small fails to do so, as shown in Figure 1(b). This experiment highlights the poor alignment capability of the training strategy used in Zero123 (Liu et al., 2023a), suggesting that its impressive NVS performance is primarily due to the large volume of training data. Furthermore, simply increasing the amount of training data does not resolve the poor alignment issue, as evidenced by the performance of Zero123-XL (Deitke et al., 2023a) in Figure 1(a). To address the misalignment issue, an additional constraint is necessary: the final generated images, after multiple denoising steps, should closely align with the GT images.

One straightforward approach to incorporate the additional constraint in latent diffusion models, as utilized by contemporary multi-view diffusion models, is to introduce a loss function such as Mean Square Error (MSE) in the VAE-encoded feature space between the generated images and the GT images. However, our extensive experiments reveal that directly applying a loss in the VAE-encoded feature space often leads to training collapse (see Section 4.5 for more details). Inspired by the recently proposed closed-loop transcription (CTRL) framework (Dai et al., 2022; Ma et al.,

2022; Tong et al., 2022), we propose to align the generated images with the GT images using the MSE of CLIP (Radford et al., 2021) **patch features**, which effectively capture fine-grained image information. We name our method Ctrl123, a CTRL-based multi-view diffusion model that significantly alleviates the misalignment problem in NVS and achieves better multi-view consistency performance. Through the lens of CTRL framework, we model the majority of current methods as broad open-loop auto-encoders (as shown in Figure 2(a)). In our work, we extend this open-loop framework to a closed-loop one by looping the generated images back into the CLIP (Radford et al., 2021) encoder (as shown in Figure 2(b)). We then optimize the latent diffusion models through minimizing the difference between the generated images and their corresponding GT images in the CLIP (Radford et al., 2021) **patch feature** space. In addition, since NVS focuses on view generation under transformed camera viewpoint, primarily through rotation, we introduce additional metrics (Average Angle (AA) and Intersection over Union (IoU)) to effectively evaluate rotation accuracy, an aspect that is often lacking in current NVS works. High values of AA and IoU indicate not only strong multi-view consistency but also excellent rotation accuracy. Through extensive experiments, we demonstrate that Ctrl123 significantly improves consistency and leads to substantially better NVS performance compared to current state-of-the-art methods. The main contributions of this paper are:

- To improve the multi-view consistency and maintain the capability of generating arbitrary novel views, we introduce Ctrl123, a novel closed-loop transcription-based multi-view diffusion model that uses CLIP (Radford et al., 2021) patch features to measure and minimize differences between generated images and the GT images, and further constrain the excessive diversity which is not beneficial for NVS.

- An in-depth experiment on a sample set of 25 training objects demonstrates that the training strategy of Ctrl123 exhibits extensively better performance on aligning with GT images than that of Zero123(Liu et al., 2023a). As shown in Table 1, Ctrl123-small improves the NVS performance by a **7 point** increase in PSNR and a **0.06 point** increase in SSIM. Furthermore, Ctrl123-small significantly improves the rotation accuracy with a **35.1%** increase in $AA^{15°}$ and **42.5%** increase in $IoU^{0.7}$ (see Table 1).

- We further train Ctrl123 on a large-scale 3D dataset Objaverse (Deitke et al., 2023b) and observe similar improvements. Ctrl123 improves PSNR, SSIM, AA and IoU on three evaluation datasets, which proves the substantially better NVS performance and rotation accuracy compared to current state-of-the-art methods (see Table 2).

## 2 RELATED WORKS

**Diffusion Models**   Diffusion models (Ho et al., 2020) demonstrate remarkable capability in image generation, especially text-to-image (Ramesh et al., 2022; Saharia et al., 2022; Nichol et al., 2021; Rombach et al., 2022). Recent studies have explored distilling knowledge from pre-trained diffusion models for text-to-3D generation through optimizing a differentiable 3D representation with image priors (Poole et al., 2022; Wang et al., 2023a; Chen et al., 2023; Lin et al., 2023; Wang et al., 2023c; Huang et al., 2023). However, this line of work suffers from optimization efficiency, resulting in blurriness and the Janus problem in the generated 3D models. This arises due to the lack of 3D-awareness in pre-trained text-to-image models.

**Multi-view Diffusion Models**   To achieve Novel view synthesis (NVS), Multi-view Diffusion Models are required to generate an object's unobserved geometry and texture, which is a prerequisite capability for 3D generation. Recently, Zero123 (Liu et al., 2023a) proposes to perform zero-shot open-set NVS by fine-tuning a pre-trained text-to-image diffusion model on multi-view renderings of diverse 3D data (Deitke et al., 2023b). However, Zero123 (Liu et al., 2023a) often generates inconsistent multi-view images. Subsequent efforts (Liu et al., 2023b; Weng et al., 2023; Voleti et al., 2024; Zheng & Vedaldi, 2024; Long et al., 2023; Yang et al., 2024) aim to resolve view inconsistency by facilitating information propagation across different views which naturally require generating multiple novel views concurrently. While these models excel in generating more consistent multi-view novel views, they alter the NVS task settings of Zero123 and face limitations in generalizing arbitrary camera poses.

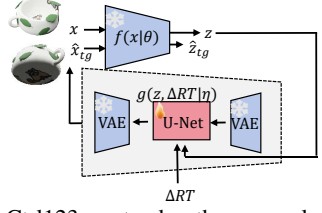

(a) Existing multi-view diffusion models (Zero123 (Liu et al., 2023a) and etc.) can be broadly viewed as an (open-loop) auto-encoder.

(b) Ctrl123 extends the open-loop multi-view diffusion models to a closed-loop framework (transcription).

Figure 2: Comparison between the training pipeline of current open-loop multi-view diffusion models and closed-loop Ctrl123.

**Reconstruction Models** Different from NeuS (Wang et al., 2023b) algorithm, Reconstruction Models (Li et al., 2023; Xu et al., 2024; Tang et al., 2024; Liu et al., 2024; Wu et al., 2024) take multi-view images generated by multi-view diffusion models as input, and generate textured 3D meshes with high quality. However, the input multi-view images of reconstruction models have different settings. Therefore, multi-view diffusion models which only generate fixed views cannot adapt to all reconstruction models, which highlight the importance of generating arbitrary novel views. While InstantMesh (Xu et al., 2024) adopts 6 fixed multi-view images generated by Zero123++ (Shi et al., 2023a), Cycle3D (Tang et al., 2024) randomly samples 4 views with elevation angles in the range [-5°, 5°] as input. Our proposed Ctrl123 can generate arbitrary consistent multiple views and can be used in all reconstruction models.

## 3 METHOD

Our goal is to learn a multi-view diffusion model that alleviates the multi-view inconsistency problem and the excessive diversity by employing the idea of closed-loop transcription. To this end, we formulate the current state-of-the-art (SOTA) methods for single-image NVS as a broad open-loop (without looping generated images in the encoder) auto-encoder framework (Section 3.1 and Figure 2(a)). Then, we extend the framework, from open-loop to closed-loop (looping generated images in the encoder), to optimize the latent diffusion model with the help of the closed-loop transcription (CTRL) framework (Section 3.2 and Figure 2(b)). Finally, the choice of training strategy is discussed in Section 3.2.

### 3.1 CURRENT MULTI-VIEW DIFFUSION MODELS AND THEIR CAVEATS

**The Formulation of Current multi-view diffusion models.** Given a single RGB image (reference image) of an object, the goal of multi-view diffusion models is to synthesize an image of the object from a different camera viewpoint (target view) given the relative camera transformation. Although various works improve their NVS performance in different aspects, we could unify their formulation as a broad open-loop auto-encoder as follows.

Suppose the random variables $\boldsymbol{X}$, $\Delta \boldsymbol{RT}$, and $\boldsymbol{X}_{tg}$ denote the reference image, the relative camera extrinsic, and the target image, respectively. The dataset $\mathcal{D}$ comprising $n$ triplets represented as $\mathcal{D} = \left\{ \left( \boldsymbol{X}^i, \Delta \boldsymbol{RT}^i, \boldsymbol{X}_{tg}^i \right) \right\}_{i=1}^n$, where the given $n$ i.i.d. samples $\boldsymbol{X}^1, \ldots, \boldsymbol{X}^n \sim \boldsymbol{X}$, $\Delta \boldsymbol{RT}^1, \ldots, \Delta \boldsymbol{RT}^n \sim \Delta \boldsymbol{RT}$, and $\boldsymbol{X}_{tg}^1, \ldots, \boldsymbol{X}_{tg}^n \sim \boldsymbol{X}_{tg}$. Specifically, $\Delta \boldsymbol{RT}^i = (\Delta \boldsymbol{R}^i, \Delta \boldsymbol{T}^i)$, where $\Delta \boldsymbol{R}^i \in \mathbb{R}^{4 \times 3}$ and $\Delta \boldsymbol{T}^i \in \mathbb{R}^4$ respectively represent the relative camera rotation and translation. Current methods aim to learn a model $h$ that synthesizes new image $\hat{\boldsymbol{X}}_{tg}$ from the reference image $\boldsymbol{X}$ under the relative camera transformation $\Delta \boldsymbol{RT}$, i.e,

$$\hat{\boldsymbol{X}}_{tg} = h(\boldsymbol{X}, \Delta \boldsymbol{RT}), \tag{1}$$

where $\hat{\boldsymbol{X}}_{tg}$ is the synthesized target view image. The goal is for $\hat{\boldsymbol{X}}_{tg}$ to be perceptually similar to the GT target view $\boldsymbol{X}_{tg}$.

The model $h$ in current methods is composited by an encoder $f : \boldsymbol{X} \mapsto \boldsymbol{Z}$, parameterized by $\theta$, and a decoder $g : (\boldsymbol{Z}, \Delta \boldsymbol{RT}) \mapsto \hat{\boldsymbol{X}}_{tg}$, parameterized by $\eta$, i.e., $h(\boldsymbol{X}, \Delta \boldsymbol{RT}) = g(f(\boldsymbol{X}), \Delta \boldsymbol{RT})$. Specifically, the encoder $f$ is commonly chosen to be the ViT (Dosovitskiy et al., 2020) model of a pre-trained CLIP (Radford et al., 2021) and kept frozen during the training. Let $\boldsymbol{Z} = f(\boldsymbol{X}; \theta) \in \mathbb{R}^d$, where $\boldsymbol{Z}$ is the CLIP (Radford et al., 2021) class feature. The decoder $g$ is commonly implemented as a process of multi-step conditional latent denoising (Liu et al., 2023a) with VAE encoder/decoder

(as shown in the dashed box of Figure 2(a)). In Zero123 (Liu et al., 2023a), the process is conditioned on a transformed feature $e = \psi(Z, \Delta RT)$ conditioned on the feature $Z$ and relative camera extrinsic $\Delta RT$, where $\psi$ is a linear layer. Then, for step $t \in \{0, \dots, T\}$, the noise $\hat{\epsilon}$ is predicted, using an U-net $\phi$, based on the predicted target view image at time step $t$, and the condition $e$, i.e., $\hat{\epsilon} = \phi(\hat{x}_{tg,t}, e, t)$. $\hat{x}_{tg,t}$ is the latent target view feature $x_{tg}$ plus $t$ steps noise where $x_{tg}$ is latent feature of target view image $X_{tg}$ by VAE encoder.

The multi-step denoising is performed through a denoiser $\mathcal{S}(\cdot)$, based on denoising diffusion implicit models (DDIM) (Song et al., 2020). The predicted target view image at time step $t - 1$ is generated by the denoiser, i.e., $\hat{x}_{tg,t-1} = \mathcal{S}(\hat{x}_{tg,t}, \hat{\epsilon}, t)$ [1]. For the initial step $t = t_\infty$, the $\hat{x}_{tg,t_\infty}$ is randomly sampled from isotropic Gaussian. Then, after $t_\infty$ denoising steps, we get $\hat{x}_{tg,0} = \hat{x}_{tg}$. Finally, the denoised latent feature $\hat{x}_{tg}$ was lifted to generate target view image $\hat{X}_{tg}$ through the VAE decoder.

We denote random variable $\hat{X}_{tg} \doteq g(Z, \Delta RT)$ as the generated images decoded from Gaussian noise $\hat{x}_{tg,t_\infty}$ through $t_\infty$ denoising steps according to $\mathcal{S}$ conditioned on $Z$ and $\Delta RT$. Current multi-view diffusion models can be summarized as the following open-loop framework,

$$X \xrightarrow{f(X;\theta)} Z \xrightarrow{g(Z,\Delta RT;\eta)} \hat{X}_{tg}. \tag{2}$$

Because VAE encoder/decoder is frozen during training, we denote the trainable parameters in $g$ as $\eta^*$. Such multi-view diffusion models typically adopt a standard diffusion training strategy by supervising single-step denoising results with a score-matching loss, i.e.,

$$\min_{\eta^*} \mathbb{E}\|\phi(\hat{x}_{tg,t}, \psi(Z, \Delta RT), t) - \epsilon\|_2^2, \tag{3}$$

where the expectation is taken over the encoded feature $Z$, timestep $t$, relative camera extrinsic $\Delta RT$, and randomly sampled Gaussian noise $\epsilon$.

**The Caveats of Current NVS Methods.** While the showcased results in current works appear impressive, we make a critical observation previously overlooked: when evaluated on the training data, the generated images of novel views often lack alignment with the GT images, as exemplified in Figure 1(a). This misalignment problem is rooted in the excessive diversity caused by the traditional training strategy which separately trains the denoiser at different noise levels using solely score matching loss without any constraints on the generated images obtained through the whole denoising process. The above claim is demonstrated by the results of over-fitting experiments, as shown in Figure 1(b) and described in Section 4.2. The excessive diversity of the generated images is not beneficial for NVS and results in multi-view inconsistency problem. Therefore, in this work, we explore ways to enforce alignment between the generated images and GT images to learn the suitable diversity for the NVS task.

### 3.2 CTRL123: CONSISTENCY VIA A CLOSED-LOOP FRAMEWORK

A straightforward method to enforce sample-wise alignment is to add a loss such as Mean Square Error (MSE) in the VAE-encoded **feature space** between the generated images $\hat{X}_{tg}$ and GT images $X_{tg}$. However, through extensive experiments, a direct loss in the VAE-encoded feature space often suffers training difficulties leading to training collapse (See according ablation study in Table 8).

The recently proposed **closed-loop transcription** (CTRL) framework (Dai et al., 2022) offers a promising solution to this problem. In the CTRL framework, $X$ represents the input image and $\hat{X}$ represents the reconstructed image within an auto-encoder. The difference between $X$ and $\hat{X}$ can be measured through the distance between their corresponding features $Z = f(X;\theta)$ and $\hat{Z} = f(\hat{X};\theta)$ mapped through the same encoder, i.e.,

$$X \xrightarrow{f(X;\theta)} Z \xrightarrow{g(Z;\eta)} \hat{X} \xrightarrow{f(\hat{X};\theta)} \hat{Z}. \tag{4}$$

Inspired by CTRL, we apply this idea to the current multi-view diffusion model that measures the difference between generated views and GT target views in the CLIP feature space. In other words,

---

[1] $\mathcal{S}(\hat{x}_{tg,t}, \hat{\epsilon}, t) = \frac{1}{\sqrt{\bar{\alpha}_t}}(\hat{x}_{tg,t} - \frac{1-\alpha_t}{\sqrt{1-\bar{\alpha}_t}}\hat{\epsilon})$ where $\bar{\alpha}_t = \prod_{i=1}^{t} \alpha_i$ and $\alpha_t = 1 - \beta_t$, $\beta_t$ is a pre-defined variance of $t$-th step.

we apply it to framework 2 to get the following new framework:

$$X \xrightarrow{f(X;\theta)} Z \xrightarrow{g(Z,\Delta RT;\eta)} \hat{X}_{tg} \xrightarrow{f(\hat{X}_{tg};\theta)} \hat{Z}_{tg}. \tag{5}$$

Different from the direct CTRL formulation equation 4, we can not directly calculate the loss between $Z$ and $\hat{Z}_{tg}$ since they are the features of different views. Hence, we add the feature $Z_{tg}$ of the GT target view $X_{tg}$ in the framework 5 as the following,

$$X \xrightarrow{f(X;\theta)} Z \xrightarrow{g(Z,\Delta RT;\eta)} \hat{X}_{tg} \xrightarrow{f(\hat{X}_{tg};\theta)} \hat{Z}_{tg}, \tag{6}$$
$$X_{tg} \xrightarrow{f(X_{tg};\theta)} Z_{tg}.$$

Now we can measure the difference between $Z_{tg}$ and $\hat{Z}_{tg}$ through Mean Squared Error,

$$\min_{\eta^*} \|Z_{tg} - \hat{Z}_{tg}\|_2^2. \tag{7}$$

Building upon the previously established definition of $Z$, we identify it as the class feature within the outputs of the Vision Transformer (ViT) model. Revisiting the ViT model, its outputs are bifurcated into class features and patch features. The class features capture high-level information, whereas the patch features are indicative of low-level information. Our methodology prioritizes the analysis of patch features due to their richer and more detailed informational content, which significantly aligns with GT images and enhances consistency. This preference is empirically validated in our ablation study, which demonstrates superior performance of patch features over class features (refer to Table 8 for detailed results).

To accommodate this distinction, we introduce a revised notation for the encoder outputs, denoted as $[Z_c, Z_p] = f(X, \theta)$ and $[Z_{tg,c}, Z_{tg,p}] = f(X_{tg}, \theta)$, where $Z_c$ and $Z_{tg,c}$ represent the high-level class features, and $Z_p$ and $Z_{tg,p}$ correspond to the low-level patch features. Re-writing the framework 6 with the re-defined notation as the following,

$$X \xrightarrow{f(X;\theta)} Z_c \xrightarrow{g(Z_c,\Delta RT;\eta)} \hat{X}_{tg} \xrightarrow{f(\hat{X}_{tg};\theta)} \hat{Z}_{tg,p} \tag{8}$$
$$X_{tg} \xrightarrow{f(X_{tg};\theta)} Z_{tg,p},$$

where $Z_c$ is the class feature of reference view images $X$ and $Z_{tg,p}$, $\hat{Z}_{tg,p}$ are the patch features of target view ground truth images $X_{tg}$ and generated target view images $\hat{X}_{tg}$, and we can measure the difference between $Z_{tg,p}$ and $\hat{Z}_{tg,p}$ through Mean Squared Error,

$$\min_{\eta^*} \|Z_{tg,p} - \hat{Z}_{tg,p}\|_2^2. \tag{9}$$

**Alternative Training Strategy**  Instead of directly minimizing the closed-loop loss 9 and the score-matching loss, we propose an alternative training strategy which is ablated in Table 7. We perform fine-tuning on pre-trained Zero123 (Liu et al., 2023a). One round of training involves 1) $m$ iterations of closed-loop training via CTRL framework as depicted in 8 with supervision loss according to Equation 9. We call this process CL training; 2) $n$ iterations of standard diffusion model fine-tuning with score-matching loss. We call this process SM training for short. Such an alternative training strategy is more efficient and practical as the hyper-parameters of CL training and SM training are vastly different. It is time-consuming and energy-inefficient to grid search a sweet spot of hyper-parameters for both CL and SM training. In section 4.5 we demonstrate the effectiveness of this alternative training strategy.

## 4  EXPERIMENTS

We evaluate Ctrl123 on the task of NVS and show promising results of 3D reconstruction. 3D reconstruction is a challenging task that requires strong multi-view consistency. In Section 4.1, we introduce the metrics, Angle Accuracy (AA) and Intersection over Union (IoU), used to additionally measure rotation accuracy of the generated images. In Section 4.2, we provide results on a small dataset which only obtains 25 objects to compare the alignment capability through a simple over-fitting task, and, in Section 4.3, provide results on large-scale datasets to compare the NVS performance on evaluation datasets. Results on 3D reconstruction are provided in Section 4.4. Ablations on the training strategy choices are presented in Section 4.5.

Table 1: Quantitative comparison between generated images and GT images when trained on the small dataset, measured with 4 metrics: SSIM(↑), PSNR(↑), $AA^{15°}$(↑), and $IoU^{0.7}$(↑). As shown in results of over-fitting performance, Ctrl123's training strategy shows better alignment capability.

| Method vs. Metrics | NVS Performance | Rotation Accuracy |
|---|---|---|
| Zero123-small | SSIM:0.8254 PSNR:19.0839 | $AA^{15°}$:22.62% $IoU^{0.7}$:30.93% |
| Ctrl123-small (1 round) | SSIM:0.8687 PSNR:23.6080 | $AA^{15°}$:32.02% $IoU^{0.7}$:55.01% |
| Ctrl123-small (2 rounds) | SSIM:**0.8867** PSNR:**26.5348** | $AA^{15°}$:**57.78%** $IoU^{0.7}$:**73.44%** |

Table 2: Quantitative comparison of NVS performance between Ctrl123 and five baselines, measured with four different metrics: PSNR (↑), SSIM (↑), $AA^{15°}$(↑), and $IoU^{0.7}$(↑), and evaluated on 3 datasets: GSO (Downs et al., 2022), RTMV (Tremblay et al., 2022), and OmniObject3D (Wu et al., 2023). ∗ means the model shows different performance from that reported in its literature, because we evaluate under different settings.

| Method | GSO (Downs et al., 2022) | | | | RTMV (Tremblay et al., 2022) | | | | OmniObject3D (Wu et al., 2023) | | | |
|---|---|---|---|---|---|---|---|---|---|---|---|---|
| | PSNR | SSIM | $AA^{15°}$ | $IoU^{0.7}$ | PSNR | SSIM | $AA^{15°}$ | $IoU^{0.7}$ | PSNR | SSIM | $AA^{15°}$ | $IoU^{0.7}$ |
| Zero123 | 19.4263 | 0.8418 | 14.41% | 47.32% | 9.7123 | 0.4919 | 8.04% | 8.91% | 16.7038 | 0.7805 | 14.56% | 48.29% |
| SyncDreamer∗ | 14.9284 | 0.7911 | 10.68% | 32.29% | 6.1244 | 0.3594 | 3.89% | 2.06% | 17.0247 | 0.7917 | 10.28% | 31.89% |
| Consistent123 | 20.1710 | 0.8544 | 16.27% | 59.73% | 10.5970 | 0.5384 | 10.63% | 11.16% | 17.1215 | 0.7768 | 16.59% | 60.28% |
| SV3D∗ | 18.1537 | 0.8157 | 15.97% | 56.83% | 8.4416 | 0.4334 | 9.60% | 10.81% | 17.0651 | 0.7932 | 13.26% | 57.19% |
| Free3d∗ | 19.3164 | 0.8480 | 15.42% | 55.66% | 9.8660 | 0.5153 | 9.46% | 10.69% | 17.9823 | 0.8018 | 15.29% | 55.18% |
| Ctrl123 | **20.6336** | **0.8689** | **16.96%** | **62.11%** | **10.8938** | **0.5684** | **11.98%** | **12.41%** | **18.3437** | **0.8279** | **17.34%** | **63.49%** |

**Settings.** We use Zero123 (Liu et al., 2023a), SyncDreamer(Liu et al., 2023b), SV3D (Voleti et al., 2024), Free3D (Zheng & Vedaldi, 2024) and Consistent123 (Weng et al., 2023) as our baselines, and train Ctrl123 on the public 3D dataset - Objaverse (Deitke et al., 2023b) which contains around 800K diverse 3D models created by artists. See appendix A for the implementation details and discussions about efficiency issue.

## 4.1 ANGLE ACCURACY (AA) AND INTERSECTION OVER UNION (IOU)

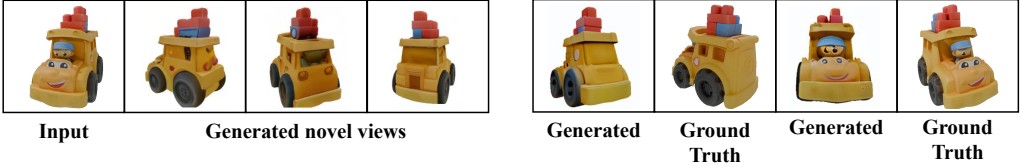

| **Input** | **Generated novel views** | | | **Generated** | **Ground Truth** | **Generated** | **Ground Truth** |

(a) **Multi-view inconsistency**: inconsistency happens at the back of car on generated multiple views.

(b) **Bad rotation accuracy**: Even if the quality of generated images is good, the rotation is not accurate.

Figure 3: The performance of NVS can be evaluated based on two key aspects: **multi-view consistency**, which refers to whether generated multiple views are consistent among each other, and **rotation accuracy**, which refers to whether the rotation achieved by the multi-view diffusion models is correct.

The performance of multi-view diffusion models can be evaluated based on two key aspects: **multi-view consistency** and **rotation accuracy**, as illustrated in Figure 3. Current works (Shi et al., 2023a; Liu et al., 2023b; Shi et al., 2023b; Long et al., 2023; Zheng & Vedaldi, 2024; Voleti et al., 2024; Yang et al., 2024) assess model performance using traditional image generation metrics such as KID, FID, PSNR, and SSIM, which effectively measure multi-view consistency. However, these traditional metrics fall short in evaluating rotation accuracy, particularly in the presence of slight and reasonable artifacts. In our work, we introduce two new metrics—**Angle Accuracy (AA)** and **Intersection over Union (IoU)**—to assess rotation accuracy at both **fine-grained** and **coarse** levels. It is important to note that AA is not designed to disentangle the effects of artifacts but rather to additionally focus on rotation accuracy. Therefore, high AA indicates strong multi-view consistency and rotation accuracy. Furthermore, If the multi-view consistency is significantly poor, evaluating rotation accuracy becomes meaningless.

**Angle Accuracy (AA).** AA is first introduced in (Zhou et al., 2019). In our work, we utilize MegaPose (Labbé et al., 2023), an excellent single-image pose estimator, to predict the camera pose

based on the generated images and the GT meshes. We then calculate the angular difference between the predicted camera pose and the GT camera pose. We define accurate rotation as the situation that the angular difference falls within a predefined threshold of $x°$, and calculate the percentage of accurate rotation, denoted as $AA^{x°}$. As shown in Appendix D, we set $x$ to 5, 10, 15, and 20 degrees and calculate the corresponding AA, all of which demonstrate the superior rotation accuracy of Ctrl123. Additionally, as presented in Appendix D, we also provide the results of MegaPose (Labbé et al., 2023) on generated images, which confirm that MegaPose can accurately predict camera poses even in the presence of slight artifacts. To account for reasonable artifacts, we select $AA^{15°}$ as the final metric reported in the comparison tables (see more details in Appendix D about the credibility of the selection of $15°$ as the threshold).

**Intersection over Union (IoU).**    We also utilize the segmentation metric IoU (Garcia-Garcia et al., 2017) to evaluate a more coarse aspect of rotation accuracy. DIS (Qin et al., 2022) is employed to predict the masks of both the generated images and the GT images, which are then used to calculate the IoU. The IoU is defined as the ratio of the overlap between the predicted masks and GT masks to their union. We define accurate rotation as the condition that the IoU falls within a predefined threshold of $x$, and calculate the percentage of accurate rotations, denoted as $IoU^x$.

For all experiments in this paper, we report SSIM($\uparrow$), PSNR($\uparrow$), $AA^{15°}(\uparrow)$ and $IoU^{0.7}(\uparrow)$ to compare the multi-view consistency and rotation accuracy of the generated images among all methods. ($\uparrow$) denotes the higher the better, verse vice for ($\downarrow$)).

## 4.2 Improved Alignment Abilities with Ground Truth Images

To evaluate the alignment capability with GT images, we collect a small dataset from Objaverse (Deitke et al., 2023b), consisting of 25 objects, each with 12 randomly sampled views. Our goal was to assess the alignment capability of Zero123's (Liu et al., 2023a) training strategy and Ctrl123's training strategy by evaluating their over-fitting performance on this small training dataset. For clarity, we refer to them in this experiment as Zero123-small and Ctrl123-small. Regarding the experiment details, we train Zero123-small from stable diffusion for 20000 steps with a batch size of 72 (equivalent to 400 epochs). Then, starting from the checkpoint of Zero123-small at 10000 iterations, we trained Ctrl123-small for 500 closed-loop iterations (with a batch size of 12) followed by 4500 score-matching iterations (with a batch size of 72) per round.

As shown in Figure 1(b), Zero123-small fails to achieve this simple task of over-fitting, whereas Ctrl123-small succeeds. In addition, we also increased the training steps and batch size of Zero123-small, but the results remain unchanged. As demonstrated in Table 1, Ctrl123-small effectively addresses this issue, significantly outperforming Zero123-small both qualitatively and quantitatively (more quantitative results can be found in Figure 8). The PSNR improves by 7.2 points, which is nearly a $2\times$ enhancement over the baseline [2]. On a fine-grained level, the Average Angle (AA) increases from 22.62% to 57.78%. On a coarse level, the Intersection over Union (IoU) increases from 30.93% to 73.44%. Therefore, the results of this experiment demonstrate the poor alignment performance of Zero123's (Liu et al., 2023a) training strategy and confirm that the training strategy of Ctrl123 successfully resolves this issue. For further supporting the better alignment ability of Ctrl123, we also evaluate the alignment performance when training Ctrl123 on the large dataset (Deitke et al., 2023b). As shown in Table 3, we provide the quantitative results on sampled views of randomly selected 100 objects in the large trainset (Deitke et al., 2023b), which also proves the better alignment capability of Ctrl123.

## 4.3 Comeparison with SOTA

We evaluate the NVS performance of Ctrl123 and compare it with five other NVS models using the task settings defined by Zero123 (Liu et al., 2023a). The evaluation is conducted on the GSO (Downs et al., 2022), OmniObject3D (Wu et al., 2023), and RTMV (Tremblay et al., 2022) datasets, as shown in Table 2. Our evaluation focuses on two key aspects: 1) capability of generating images of arbitrary novel views, and 2) the requirement that multi-view diffusion models are conditioned only on

---

[2]Based on the definition of PSNR, a 3-point increase equates to a $1\times$ visual quality improvement (Wang et al., 2004).

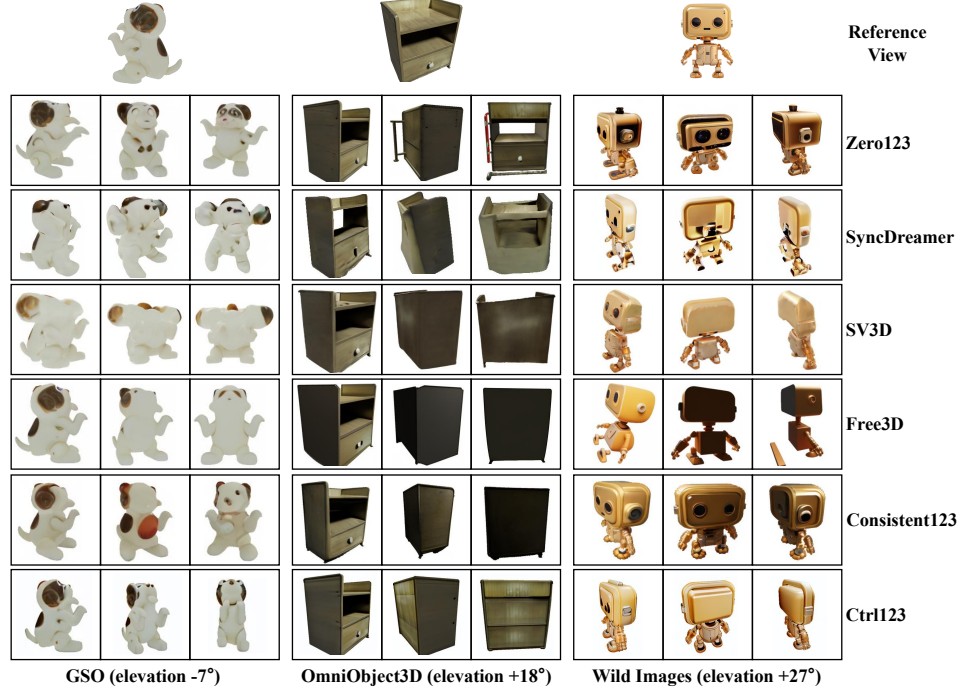

Figure 4: Qualitative comparison of NVS generalization capability on GSO (Downs et al., 2022), OmniObject3D (Wu et al., 2023) and wild images after training on the Objaverse dataset. Ctrl123 shows better generation quality and multi-view consistency than other works.

a single image and relative camera transformation. To address the first aspect, we select 20 objects from different categories and randomly sampled 200 triplets $\left(X^i, \Delta RT^i, X^i_{tg}\right)$ per object for each dataset. For the second aspect, we evaluate all models conditioned on a single image and relative camera transformation. Since SyncDreamer (Liu et al., 2023b), SV3D (Voleti et al., 2024), and Free3D (Zheng & Vedaldi, 2024) were originally evaluated using different settings, we re-evaluate them under our settings, as indicated in Table 2 as SyncDreamer*, SV3D*, and Free3D*. Sync-Dreamer (Liu et al., 2023b) only generates images with a fixed elevation of $30°$, SV3D (Voleti et al., 2024) performs bad when some kinds of relative elevation transformation is introduced (as demonstrated in Appendix B.1), and Free3D (Zheng & Vedaldi, 2024) performs well only when provided with additional absolute camera extrinsic (as demonstrated in Appendix B.2). Therefore, when tasked with generating arbitrary novel views conditioned solely on single image and relative camera transformations, these models do not perform well. Additionally, Zero123++ (Shi et al., 2023a) is excluded from our comparison because it cannot generate arbitrary views, Wonder3D (Long et al., 2023) and Hi3D (Yang et al., 2024) are also excluded because: Wonder3D is trained on RGB-D images and only generates 6 fixed views, Hi3D cannot deal with the relative elevation transformation.

As shown in Table 2, Ctrl123 outperforms other models across all four metrics (PSNR, SSIM, $AA^{15°}$, and $IoU^{0.7}$) on the evaluation datasets. We provide a qualitative comparison in Figure 4, visualizing the NVS differences on GSO (Downs et al., 2022), OmniObject3D (Wu et al., 2023), and wild images between Ctrl123 and other models (see more comparisons on these datasets and RTMV (Tremblay et al., 2022) in Appendix C.2). As shown in Figure 4, under our evaluation settings, SV3D* shows multi-view consistency but low generation quality (the wild images case), and Free3D* and Consistent123 show the lack of texture when generating the back views (the OmniObject3D case). In addition, even on some difficult cases (like the GSO case in Figure 4), Ctrl123 still outperforms other multi-view diffusion models.

## 4.4 3D RECONSTRUCTION

With the better consistency between multi-view images shown in Figure 4 and the capability of generating arbitrary novel views, we can utilize any reconstruction models to reconstruct 3D meshes with better quality. To further support the contribution of our work, we present the 3D reconstruction results through InstantMesh (Xu et al., 2024) in Figure 5.

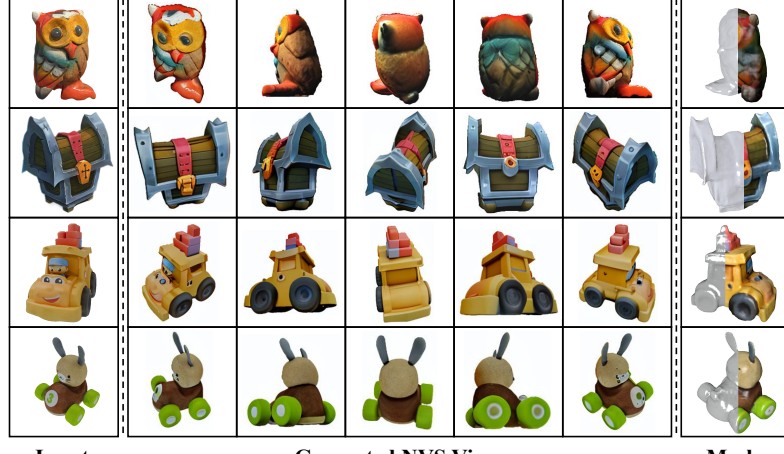

**Input**              **Generated NVS Views**              **Mesh**

Figure 5: Qualitative results of 3D reconstruction. During the 3D reconstruction, we utilize InstantMesh (Xu et al., 2024) and replace its multi-view diffusion model with Ctrl123.

### 4.5 ABLATION STUDY

All ablation studies are conducted through 1 round training with the same settings (batch size, learning rate, training steps and so on).

**Mean-square error on different space.** Applying MSE in the VAE-encoded feature space is a straightforward approach to address the misalignment issue in latent diffusion models. As shown in the ablation study presented in Appendix E, Figure 12 and Table 8, under the same training settings, we optimize our model only with MSE in pixel space and three different feature spaces: the VAE-encoded feature space, the CLIP class feature space, and the CLIP patch feature space. Our findings reveal that directly applying MSE in the VAE-encoded feature space and pixel space often results in training collapse and the looping strategy is necessary. Furthermore, due to the richer low-level detail in CLIP patch features compared to CLIP class features, models trained with MSE in the CLIP patch feature space demonstrate superior NVS performance. In addition, importantly, the MSE on different space is only conducted on generated images instead of noised images in standard training procedure, because MSE on noised images is the same as MSE on predicting noise.

**Different Training Strategy.** The "Simultaneous" training strategy, as mentioned in Section 3.2, involves adding Equation 9 and the score-matching loss of Zero123 (Liu et al., 2023a) at the same time. Because of large difference of loss values and fit race between them, it is time-consuming and energy-inefficient to grid search a sweet spot of hyper-parameters to make "Simultaneous" training strategy work. Even if we try different weights of the two loss and learning rates and report the best of them, the results in Appendix E and Table 7 show the "Alternative" strategy is better.

**Denoise Scheduler.** The number of denoising steps $t_\infty$ used in the function $g$ to generate $\hat{X}_{tg}$ is crucial for balancing the quality of $\hat{X}_{tg}$ against memory consumption. To investigate this, we conducted experiments with different numbers of denoising steps (1, 10, 30, 50) to evaluate their impact on NVS performance and rotation accuracy (see more details in Appendix E and Table 6). Ultimately, we select 50 denoising steps during CL training.

### 5 CONCLUSION

In this paper, we introduce Ctrl123, a closed-loop transcription-based multi-view diffusion method that significantly alleviates the problem of multi-view inconsistency under the task of generating arbitrary novel views. To quantitatively measure the improved NVS performance of Ctrl123, we introduce metrics AA and IoU. Through extensive experiments, we show that the closed-loop Ctrl123 significantly improves consistency performance, and leads to excellent 3D reconstruction results compared to the current SOTA methods. Note that in (Ma et al., 2022; Dai et al., 2022) the closed-loop framework is proposed as a general framework for ensuring consistency. Hence we believe such a closed-loop framework is a simple yet effective way to ensure consistency in content generation. We will leave the application of Ctrl123 on the downstream tasks to future investigation.

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

## A    IMPLEMENTATION DETAILS OF CTRL123

We train Ctrl123 on Objaverse (Deitke et al., 2023b) dataset for 1 round of alternative training. We initialize Ctrl123 with pre-trained weights of Zero123 (Liu et al., 2023a) and train on 8 A100 GPUs with 80GB memory. For each alternative training round, we conduct 500 steps of closed-loop (CL) training with a total batch size of 320 and a learning rate of $10^{-5}$, followed by 5000 steps of score-matching (SM) training with a total batch size of 1536 and a learning rate of $10^{-4}$, each taking around 2 days. We use gradient accumulation to increase the training batch size, 20 for CL training and 1 for SM training respectively. During model training, we use the Adam optimizer (Kingma & Ba, 2014) with $\beta_1 = 0.9$ and $\beta_2 = 0.99$. In all the experiments, we train our model with the 16-bit floating point (fp16) format for efficiency. In addition, during the CL training, we use DDIM (Song et al., 2020) scheduler and set $t_\infty$ as 50.

Even if the training of Ctrl123 costs a large amount of memory and we need to set large accumulation gradient step to enlarge batch size, the convergence rate of CL training is fast. Therefore, the additional cost time of Ctrl123 is not a key limited issue.

## B    MORE QUALITATIVE RESULTS OF BASELINES

### B.1    SV3D EXHIBITS POOR NVS PERFORMANCE WHEN THE RELATIVE ELEVATION
TRANSFORMATION EXISTS.

We evaluate the NVS performance of SV3D (Voleti et al., 2024) (specifically, SV3D$^p$ with the best performance) under three different settings, all of which have the same relative azimuth transformation but different relative elevation transformation ($+30°$, $+0°$ and $-30°$), as shown in Figure 6. When the relative elevation transformation is zero, the NVS performance is great, as presented in its original literature. However, when relative elevation transformation is introduced, issues such as low generation quality (as seen in the third sample in the third row of Figure 6) and inaccurate rotation (as seen in the third sample in the first row of Figure 6) occur. In our evaluation setting, both the relative elevation transformation and the relative azimuth transformation are selected randomly, which is why the metrics for SV3D (Voleti et al., 2024) in Table 2 are lower than those reported in its original literature.

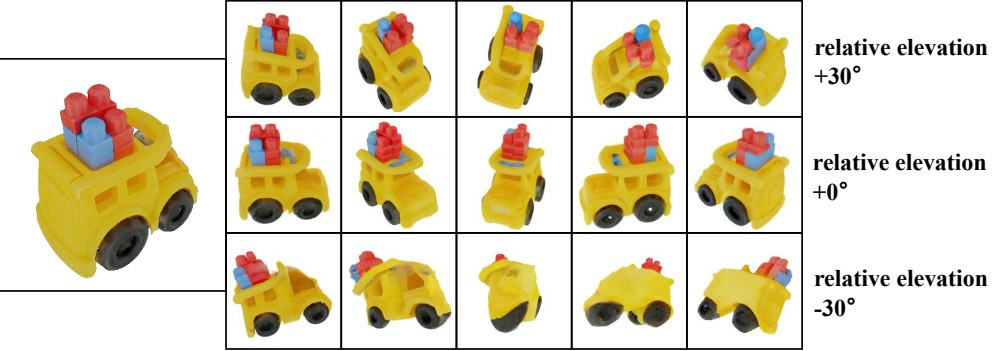

Figure 6: SV3D (Voleti et al., 2024) performs well when the relative elevation transformation is zero, but its performance degrades significantly when any relative elevation transformation is introduced. Therefore, SV3D (Voleti et al., 2024) performs bad under the task of generating arbitrary novel views.

### B.2    FREE3D PERFORMS WELL ONLY WHEN PROVIDED WITH ADDITIONAL ABSOLUTE
CAMERA POSE INFORMATION

As shown in Figure 7, Free3D (Zheng & Vedaldi, 2024) demonstrates great NVS performance when provided with the additional absolute camera extrinsic condition. However, without this additional condition, the quality of the generated images degrades, leading to a decrease in PSNR and SSIM, as indicated in Table 2. This limitation affects the performance of Free3D (Zheng & Vedaldi, 2024),

particularly on NVS tasks on wild images. In addition, despite the quality degradation seen in the second row of Figure 7, the generated images maintain great rotational accuracy, resulting in higher AA and IoU scores than Zero123 (Liu et al., 2023a), as shown in Table 2. This phenomenon also underscores the significance of the metrics we introduce.

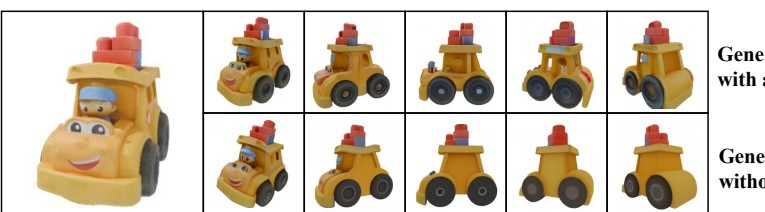

Figure 7: Free3D (Zheng & Vedaldi, 2024) performs well with the additional absolute camera extrinsic condition, but its performance degrades without this additional condition.

## C  MORE QUALITATIVE RESULTS

### C.1  MORE COMPARISON RESULTS OF THE OVER-FITTING EXPERIMENT ON THE SMALL DATASET.

In Figure 8, we showcase more cases comparing the over-fitting performance between Zero123-small and Ctrl123-small. It is evident that Ctrl123-small shows better alignment capability in the over-fitting task.

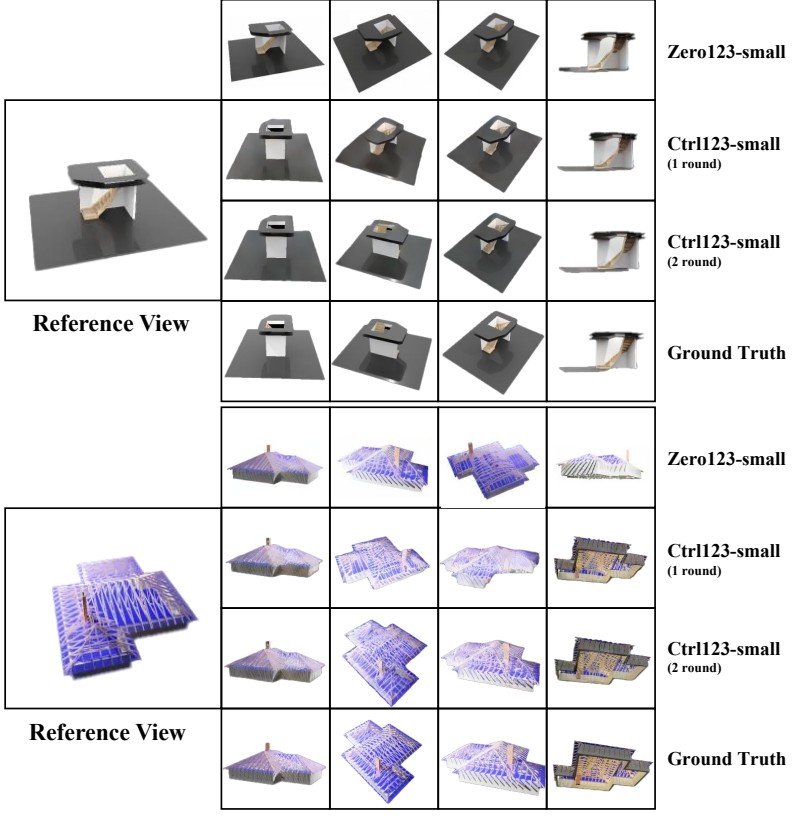

Figure 8: More qualitative comparison of the alignment performance between the generated images and the GT images trained on the small dataset (25 objects). Comparisons between Zero123-small, Ctrl123-small, and GT images.

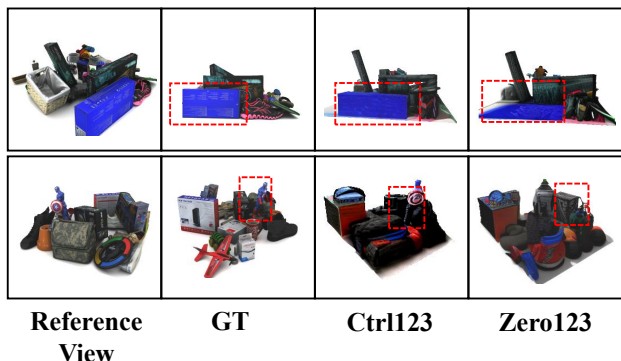

**Reference      GT      Ctrl123      Zero123**
**View**

Figure 9: Qualitative comparison of NVS generalization capability on RTMV (Tremblay et al., 2022) after training on the Objaverse (Deitke et al., 2023b) dataset.

For better support the better alignment capability of Ctrl123 in the trainset, as shown in Table 3, we also provide quantitative comparisons of Ctrl123 and Zero123 (Liu et al., 2023a) on 100 randomly selected trainig objects when trained on the large dataset (Deitke et al., 2023b), measured with four different metrics: PSNR ($\uparrow$), SSIM ($\uparrow$), AA$^{15^\circ}$($\uparrow$), and IoU$^{0.7}$($\uparrow$). The results also prove the better alignment ability of Ctrl123.

Table 3: Quantitative comparison of Ctrl123 and Zero123 on the 100 randomly selected objects from large trainset (Deitke et al., 2023b), measured with 4 different metrics: PSNR ($\uparrow$), SSIM ($\uparrow$), AA$^{x^\circ}$($\uparrow$), and IoU$^x$($\uparrow$), which proves the better alignment performance of Ctrl123.

| Method | 100 randomly selected training objects | | | |
| --- | --- | --- | --- | --- |
| | PSNR | SSIM | AA$^{15^\circ}$ | IoU$^{0.7}$ |
| Zero123 | 19.9452 | 0.8443 | 17.28% | 51.86% |
| Ctrl123 | **21.5398** | **0.8752** | **19.79%** | **65.97%** |

## C.2 MORE NVS COMPARISON RESULTS BETWEEN CTRL123 AND OTHER BASELINES.

In this subsection, we present more results of Ctrl123 after large-scale training on Objaverse (Deitke et al., 2023b) dataset. Figure 10 shows the NVS comparison, which supports the better multi-view consistency of Ctrl123. In addition, because RTMV (Tremblay et al., 2022) dataset only provide images of scenes instead of objects and all multi-view diffusion models perform bad on it, we simply provide the qualitative comparison between Ctrl123 and Zero123 in Figure 9.

# D THE CREDIBILITY OF OUR INTRODUCED AA METRIC

## D.1 PRECISE SINGLE IMAGE POSE ESTIMATION BY MEGAPOSE

To evaluate the accuracy of pose estimation through Megapose (Labbé et al., 2023), we randomly select 5 objects from GSO dataset (Downs et al., 2022) and randomly sampled 15 view pairs, ensuring that the angular difference between each pair is less than 1° but not equal to 0 (to avoid the total

Table 4: Average angular difference results through Megapose for nearly identical picture pairs.

| angular difference(through MegaPose) | view-pair1 | view-pair2 | view-pair3 | *average* |
| --- | --- | --- | --- | --- |
| 3D_Dollhouse_Sink | 3.5833° | 2.4288° | 1.0024° | 5.9619° |
| JUNGLE_HEIGHT | 2.0103° | 6.6258° | 5.2131° | 5.0780° |
| Lenovo_Yoga_2_11 | 3.8219° | 6.0171° | 2.9925° | 6.9773° |
| Shark | 5.2212° | 3.9884° | 6.0070° | 5.2961° |
| Sonny_School_Bus | 6.9927° | 4.5645° | 15.1978° | 7.4337° |

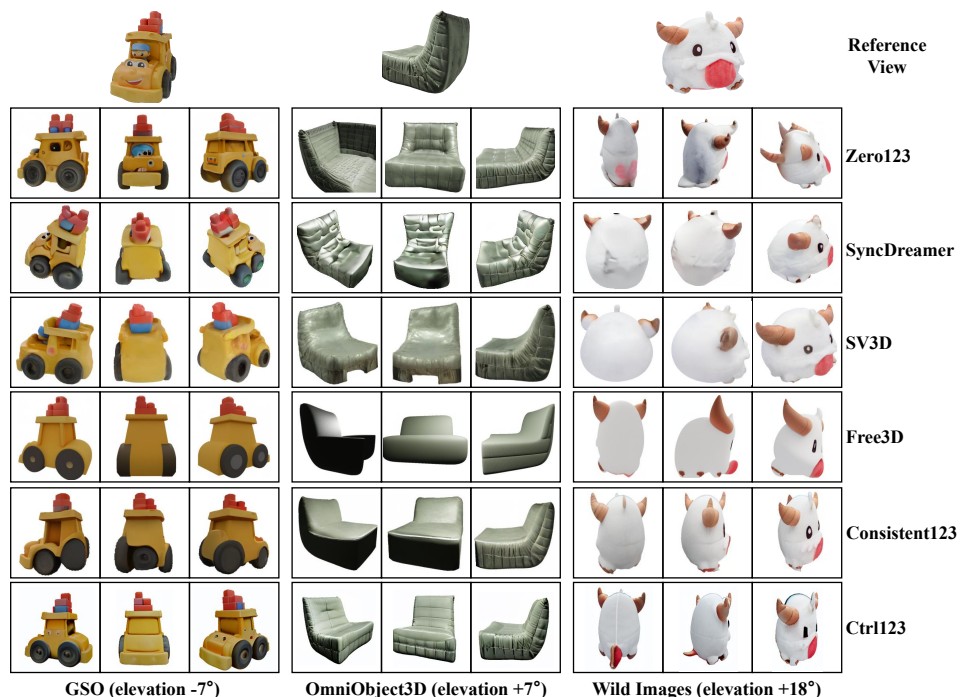

Figure 10: Qualitative comparison of NVS generalization capability on GSO (Downs et al., 2022), OmniObject3D (Wu et al., 2023) and wild images after training on the Objaverse (Deitke et al., 2023b) dataset. Ctrl123 shows better quality and consistency than other baselines.

Table 5: AA values for different thresholds of multi-rounds Ctrl123-small when trained on a small dataset.

| Method | $AA^{5°}$ | $AA^{10°}$ | $AA^{15°}$ | $AA^{20°}$ | AA(Average) |
|---|---|---|---|---|---|
| Zero123-small | 6.29% | 12.55% | 22.62% | 32.20% | 18.42% |
| Ctrl123-small (1 round) | 14.23% | 21.25% | 32.02% | 41.89% | 27.35% |
| Ctrl123-small (2 round) | **30.41%** | **39.34%** | **57.78%** | **69.51%** | **49.26%** |

same view pairs). Subsequently, we utilized MegaPose (Labbé et al., 2023) to estimate the average angular difference for each object. For brevity, Table 4 only displays results for three cases and the average results for 15 cases. In Table 4, Only one pair (view-pair3 of Sonny_School_Bus) results in an error exceeding 15°, while the average error for each object is less than 7.5° (half of 15°). These findings prove the credibility of MegaPose (Labbé et al., 2023) and our setting for the predefined threshold 15° of our introduced AA metric.

## D.2 THE PERFORMANCE OF POSE ESTIMATION ON IMAGES WITH SLIGHT ARTIFACTS BY MEGAPOSE

Because NVS models is impossible to generate images perfectly same as the ground truth images rendered from meshes, we also test the performance of MegaPose (Labbé et al., 2023) on generated images with minor artifacts and present three of them in Figure 11. It is found that the average angular difference is still below 15°, which aligns with results of Table 4, and also proves the credibility of MegaPose (Labbé et al., 2023) and our setting for the predefined threshold 15° of our introduced AA metric.

**Ground Truth Mesh Contour (Estimated Pose)**

**Rendering of Ground Truth Mesh (Estimated Pose)**

**3.21°**  **6.96°**  **4.17°**

Figure 11: Pose estimation results of MegaPose (Labbé et al., 2023) on generated images with minor artifacts. numbers below images indicate angular differences between estimated pose and ground truth pose, proving that $AA^{15^\circ}$ for assessing rotation accuracy of generated images is reliable.

### D.3 OUR SETTING FOR THE PREDEFINED THRESHOLD OF OUR PROPOSED AA METRIC

One may concern that AA of Ctrl123 is only better than that of Zero123 under a certain threshold. To eliminate this type of concern, we select 4 thresholds (5, 10, 15, and 20) and computed the corresponding AA values. Table 5 shows results for the over-fitting experiment. Results in Table 5 prove that the different selection of the threshold doesn't influence the conclusion - Ctrl123-small shows better performance than Zero123-small.

Table 6: Quantitative comparison on the number of denoising steps for $\hat{X}_{tg}$ generation.

| Method vs. Metrics | NVS Performance | Rotation Accuracy |
|---|---|---|
| Zero123 | SSIM :0.8418 PSNR :19.4263 | $AA^{15^\circ}$:14.41% $IoU^{0.7}$:47.32% |
| Ctrl123 (1 denoise steps) | SSIM:0.8294 PSNR :19.1593 | $AA^{15^\circ}$:13.56% $IoU^{0.7}$:42.18% |
| Ctrl123 (10 denoise steps) | SSIM:0.8502 PSNR:19.5739 | $AA^{15^\circ}$:15.71% $IoU^{0.7}$:48.97% |
| Ctrl123 (30 denoise steps) | SSIM:0.8595 PSNR:20.1165 | $AA^{15^\circ}$:16.47% $IoU^{0.7}$:59.32% |
| Ctrl123 (50 denoise steps) | SSIM: **0.8689** PSNR:**20.6336** | $AA^{15^\circ}$:**16.96%** $IoU^{0.7}$: **62.11%** |

## E  MORE ABLATION STUDY

Table 7: Ablation study on the optimization strategies.

| Strategy | NVS Performance | Rotation Accuracy |
|---|---|---|
| Simultaneous | SSIM:0.6218 PSNR:15.9643 | $AA^{15^\circ}$:5.21% $IoU^{0.7}$:45.41% |
| Alternative | SSIM:**0.8689** PSNR:**20.6336** | $AA^{15^\circ}$:**16.96%** $IoU^{0.7}$: **62.11%** |

**Ablation study on denoising steps when CL training.** It is slow to generate a sample with DDPM (Ho et al., 2020) by following the Markov chain of the reverse diffusion process, as $t_\infty$ can as many as a few thousand steps. One simple way to accelerate the process is to run a strided sampling schedule (DDIM) (Song et al., 2020) with every $\lceil t_\infty/n \rceil$ steps to reduce the process from $t_\infty$ to $n$ steps. The results, presented in Table 6, indicate that the NVS performance improves as the number of denoising steps increases. We choose denoising step as 50.

**Simultaneous vs Alternative training strategy.** We implement "Simultaneous" by changing the loss from the alternative version to the simultaneous version. All other settings like learning rate,

Table 8: Quantitative comparison on different types of consistency losses (Pixel space, VAE-encoded feature space, CLIP class feature space and CLIP patch feature space).

| Consistency Type | NVS Performance | Rotation Accuracy |
|---|---|---|
| Pixel | SSIM:0.6478 PSNR:14.9925 | $AA^{15°}$:3.19% $IoU^{0.7}$:38.12% |
| VAE-encoded features | SSIM:0.6518 PSNR:14.6905 | $AA^{15°}$:4.61% $IoU^{0.7}$:41.49% |
| CLIP class features | SSIM: 0.8385 PSNR:19.8953 | $AA^{15°}$:14.47% $IoU^{0.7}$:40.27% |
| CLIP patch features | SSIM:**0.8689** PSNR:**20.6336** | $AA^{15°}$:**16.96%** $IoU^{0.7}$:**62.11%** |

optimizer etc are the same. The results in Table 7 shows the "Alternative" strategy is better. Note that we also try adjusting related hyper-parameters to increase the performance of the simultaneous version training. However, all results show bad performance.

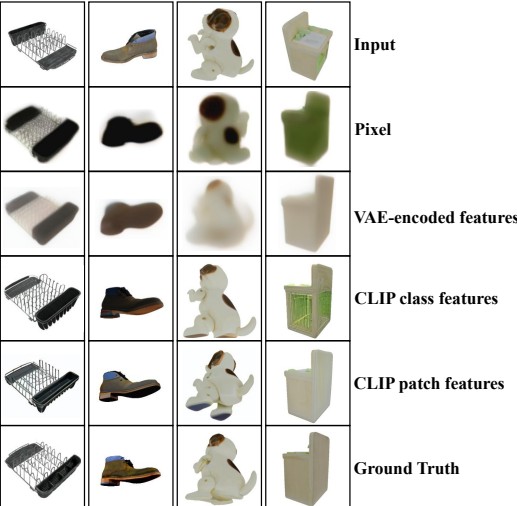

Figure 12: Qualitative comparison on the difference type of feature space for calculating Mean-Square Error loss (Pixel space, VAE-encoded feature space, CLIP class feature space and CLIP patch feature space).

**VAE-encoded feature space vs CLIP patch feature space.** To verify the effectiveness of the closed-loop framework, we conduct an ablation study that applies Mean-Square Error (MSE) on different space: pixel space, VAE-encoded feature space, CLIP class feature space and CLIP patch feature space. As shown in Table 8 and Figure 12, and enforcing MSE directly in VAE-encoded feature space and pixel space does not lead to the expected convergence and results in divergence. This highlights the challenges in achieving the additional constrain in different representation spaces and the necessity of looping generated images in the CLIP (Radford et al., 2021) encoder.

**CLIP class features vs CLIP patch features.** As discussed in section 3.2, the choice of closed-loop loss on the class feature $Z_{tg,c}$ or the patch feature $Z_{tg,p}$ is important for our methodology. We conduct the ablation study to experiment with the different choices of class features and patch features. As shown in Table 8, loss on the patch features could improve the NVS performance and rotation accuracy better than that on the class features.

