# OpenReview forum: "Ctrl123: Consistent Novel View Synthesis via Closed-Loop Transcription"
_ICLR.cc/2025/Conference — Submitted to ICLR 2025_

### Official Review · Reviewer_dbKQ · 2024-10-21

**Soundness:** 3
**Presentation:** 3
**Contribution:** 3
**Rating:** 6
**Confidence:** 4

**Summary:**

The paper consider the problem of generating novel views of an object conditioned on an input image and a camera viewpoint change. It starts from the Zero-1-to-3 formulation and adds to it an additional loss, whose purpose is to reduce the diversity of images generated by the model, with the hope to focusing on increasing multi-view consistency. This additional loss simply compares the generated and ground truth target images in the CLIP feature space after unrolling the denoising process.  The resulting new-view generator is compared to prior works using several sensible metrics and is shown to perform convincingly better.

**Strengths:**

* The method is a relatively simple tweak on Zero-1-to-3 which seems to improve NVS quality significantly. This is something that may be useful in practice.
* The use of MegaPose for assessing generated viewpoints is nice.
* The proposed tweak is simple conceptually, although it may take significant amounts of GPU memory to implement due to the unrolling.

**Weaknesses:**

* The method is a relatively simple tweak on Zero-1-to-3 which seems to improve NVS quality significantly. This is something that may be useful in practice.
* The use of MegaPose for assessing generated viewpoints is nice.
* The proposed tweak is simple conceptually (although it may take significant amounts of GPU memory to implement due to the unrolling)
* I could not find any supplementary material, and in particular no videos. This is odd for this paper, as videos are a great way for assessing qualitatively multi-view consistency.

**Questions:**

Clarity: The authors should really clarify that the function g() unrolls a large number of denoising steps. This is only clear at the very end of the paper in line 523 and is *crucial* to understand why the proposed approach makes sense. Without unrolling, there is little difference between the denoising objective used in DDPM and the proposed "closed loop" regression loss. Please consider adding this in the method description.

Likewise, the authors may want to elaborate more on why applying the regression loss after unrolling the denoiser tends to make the latter regress to the mean, killing its variance. They should also explain better how this connects to the choice of applying this loss not in VAE space or RGB space, but, further, in CLIP space, in order to "soften" variance killing and hence mode collapse.

The point of unrolling should probably be made quite evident in Figure 2 too, making it an obvious part of the diagram. This is a good place to show how one loss is applied after unrolling.

Novelty is relatively modest, but then again the paper introduces a simple tweak that does result in SoTA performance, which is nice. This is a well explored art and it is not trivial to outperform prior works with a simple change to a relatively old baseline. However, the fact that no videos are given in the supplementary material makes me wonder how well these claims hold up qualitatively. The images in the paper seem fine to me, but showing an animation can really make most inconsistencies jump out.

The authors should discuss a bit more the issue with unrolling in terms of memory impact, and do so in the main paper. 50 steps of unrolling mean that the method can multiply GPU memory fifty-fold, meaning that only ultra-tiny training batches are possible, I presume.

# Minor issues

* You should have a space (or punctuation) after citations
* Zero123 is not styled correctly; the paper is called Zero-1-to-3. You should try to match as closely as possible the styling used by the original authors. Also, it is a "to", not a "two", in the middle.
* Line 212: the "decoder" is usually stochastic, not a function. In general, I found the notation a little confusion as f() is a function, and g() is a distribution, but they are marked in the same way in the diagrams. There is no indication that g() is sampled from.
* Eqs (5) to (9): notation is essentially repeated twice, which seems unnecessary
* Fig (4): you could have included ground truth new viewpoints for reference

**Details Of Ethics Concerns:**

The proposed model is an image generator, hence potentially sensitive, but it is mostly trained on relatively unproblematic dataset. Furthermore, the specific training done here is likely to increase risks of the third-party base models.

---

### Official Review · Reviewer_NPue · 2024-10-30

**Soundness:** 1
**Presentation:** 1
**Contribution:** 1
**Rating:** 3
**Confidence:** 5

**Summary:**

This paper proposes Ctrl123, a multi-view diffusion model that improves view consistency in novel view synthesis. By enforcing alignment with ground truth images in the CLIP patch feature space, Ctrl123 reduces excessive diversity in generative models, achieving consistency across views.

**Strengths:**

The paper introduces a novel approach, Ctrl123, which uniquely combines closed-loop transcription with CLIP-based patch features to enhance multi-view consistency in novel view synthesis. By addressing the issue of excessive diversity in generative models, this method removes a significant limitation of prior approaches like Zero123, improving applicability to NVS tasks.

**Weaknesses:**

1. This article lacks true innovation; simply adding a CLIP feature loss between ground truth and the predicted image does not constitute a novel contribution, as CLIP loss has already been employed extensively in previous works. For example, [1] has used clip feature space alignment to supervise the reconstruction of the scene, which is exactly the same as your method.
2. The authors spend too much space in Sec 3.1 to explain the previous work. You should try to simplify the content in Sec 3.1 and use more space to explain your own work. In addition, your formulas 2, 4, 5, 6, and 8 are too similar. You can consider simplifying the repeated content.

[1] Jain A, Tancik M, Abbeel P. Putting nerf on a diet: Semantically consistent few-shot view synthesis[C]//Proceedings of the IEEE/CVF International Conference on Computer Vision. 2021: 5885-5894.

**Questions:**

1. I suggest that the author draw inspiration from other works on multi-view consistency to enhance the novelty of their approach. For example, [2] improves consistency by optimizing ray casting, while [3] achieves this by creating a smooth camera trajectory. More detailed innovations could be explored in the image generation process, such as adding geometric constraints, applying regularization, optimizing pixel rays, or incorporating more explicit 3D modeling to strengthen consistency, rather than simply adding a CLIP feature loss.
2. How does the author select the small dataset used to test alignment capability in Table 1? Is it representative? A similar question arises in the SOTA comparison in Section 4.3, where only 20 objects are used for quantitative analysis. Using more objects would strengthen the evidence for the method's generalization ability.
3. For 3D reconstruction, the author should include comparative experiments with other methods.

[2]	Seo S, Chang Y, Kwak N. Flipnerf: Flipped reflection rays for few-shot novel view synthesis[C]//Proceedings of the IEEE/CVF International Conference on Computer Vision. 2023: 22883-22893.
[3]	Kwak J, Dong E, Jin Y, et al. Vivid-1-to-3: Novel view synthesis with video diffusion models[C]//Proceedings of the IEEE/CVF Conference on Computer Vision and Pattern Recognition. 2024: 6775-6785.

---

### Official Review · Reviewer_vgN2 · 2024-11-02

**Soundness:** 3
**Presentation:** 1
**Contribution:** 2
**Rating:** 3
**Confidence:** 4

**Summary:**

The paper proposes a novel multi-view diffusion model named Ctrl123, which significantly improves multi-view consistency in novel view synthesis (NVS) while retaining the flexibility of generating arbitrary novel views. The method addresses the core issue of excessive diversity in generative models that leads to inconsistencies, by leveraging a closed-loop transcription-based framework that enforces alignment in the CLIP patch feature space. The main contributions include the introduction of the Ctrl123 method, improved training alignment with ground truth, and performance gains shown through experiments on training objects and a large-scale 3D dataset, Objaverse.

**Strengths:**

- Ctrl123 effectively improves multi-view consistency in novel view synthesis while maintaining the flexibility to generate arbitrary novel views. This addresses a key challenge in existing methods by balancing consistency with generation diversity.
- Also, the training strategy of Ctrl123 demonstrates improvements in quantitative metrics, such as PSNR and SSIM, as well as rotation accuracy metrics (AA and IoU). These results show that Ctrl123 not only aligns better with ground truth images but also surpasses the performance of prior models like Zero123.
- Finally, when trained on the large-scale 3D dataset Objaverse, Ctrl123 maintains its performance across different evaluation datasets, showcasing its scalability and robustness in handling diverse 3D data while improving view consistency and rotation accuracy.

**Weaknesses:**

- The biggest drawback of this paper lies in the difficulty of understanding how performance and results have improved specifically. The LPIPS metric (used in Zero123++) is not mentioned in the evaluation, and it might be worth considering if the authors could add this to their assessments. Could you include the LPIPS metric in the evaluation and comparison?

- The paper does not cite Zero123++ in the abstract, but it indirectly refers to Zero123++, by mentioning its fundamental limitation by mentioning 'restricted fixed view generation.' Does this mean the performance of Ctrl123 is better than Zero123++? However, there is no comparative analysis with Zero123++, so it is unclear what has improved in this paper. For the rebuttal process, could you present a comparative analysis with Zero123++? If there is a methodological reason for not including a comparative analysis with Zero123++, it would be helpful to explain why such a comparison was not included.

- It is stated that an in-depth analysis was conducted on a set of 25 objects. However, the current results seem to focus solely on simple, specific objects such as toys, dolls, or avatars for NVS. Can the method handle more complex objects for NVS? It seems that Zero123++ tackles more challenging problems compared to this paper. If good results are also produced for more complex sets, I suggest reflecting this in the rebuttal process.

- Ctrl123 claims to enforce alignment using CLIP, but how would it have been if the evaluation included a CLIP score? It would be beneficial to include the CLIP score in the evaluation metrics and discuss how the CLIP score relates to alignment and consistency improvements.

- Due to these points, it is difficult to objectively evaluate the results. While the claim that consistency is achieved can be acknowledged, it is challenging to give a high score due to the issue of objectively evaluating the results.

**Questions:**

Mentioned in the weaknesses section

---

### Official Review · Reviewer_Hca8 · 2024-11-04

**Soundness:** 3
**Presentation:** 2
**Contribution:** 2
**Rating:** 5
**Confidence:** 4

**Summary:**

This paper addresses the problem of object-centric novel view synthesis from a single input image. It builds upon a pretrained Zero-123 model and applies rounds of mixed closed-loop training and standard diffusion training on the Objaverse dataset. In closed-loop training, the novel view rendering loss is optimized in the CLIP feature space over 50 denoising steps. The authors first demonstrate the effectiveness of closed-loop training through an overfitting experiment on 25 objects, showing significantly improved PSNR scores. They then compare their approach with recent state-of-the-art methods on GSO, RTMV, and OmniObjects3D, showing enhanced performance.

**Strengths:**

1. The author achieves state-of-the-art results on three common benchmarks!
2. The author designs Angle Accuracy and Mask IoU to closely evaluate the 3D consistent, which is interesting.
3. The authors conduct extensive ablations on the design space of the closed-loop reconstruction loss, investigating factors such as the number of denoising steps, the feature extractor (VAE vs. CLIP), and the effectiveness of simultaneous vs. alternating training.

**Weaknesses:**

1. The proposed methods is very computationally heavy, back-propogating from 50 steps of network evaluations. Can the authors provide more details about the speed and memory cost of the proposed closed-loop training?

2. The methods sections is not writting very clearly (e.g. section 3.2)! A few concise paragraphs presenting the loss function and a diagram illustrating backpropagation through 50 denoising steps would improve readability. Since I am not familar with the Closed-Loop paper, (cited in line 258), and it takes me a long time to understand section 3.2.

3. I recommend the authors cite relevant works on alignment for diffusion models, such as "Directly Fine-Tuning Diffusion Models on Differentiable Rewards." The method in this paper is conceptually similar, with MSE in the CLIP feature space serving as a reward signal.

4. Figure-2 has formatting errors.

**Questions:**

1. Out of curiosity, what are the computational and memory costs of optimizing through the full denoising steps? Was any heavy engineering work required to make training efficient?
2. Have the authors considered using LPIPS instead of MSE in the CLIP feature space? Since GRM, LGM, GS-LRM both uses LPIPS for better visual quality. My feeling here is that the CLIP loss in this paper is similar to a LPIPS loss.
3. The authors report that alternating between closed-loop and diffusion training produces the best results. Have you tested performance when using only closed-loop training (e.g., no diffusion training after 500 steps of closed-loop training)? Additionally, what is the performance drop if closed-loop training is removed and the saved computational resources are allocated to additional diffusion training, as in the setup for Table 2?

---

### Official Review · Reviewer_4Bxn · 2024-11-04

**Soundness:** 3
**Presentation:** 1
**Contribution:** 2
**Rating:** 3
**Confidence:** 4

**Summary:**

The paper proposes a method labelled Ctrl123 that is an extension of the Zero123 model for new view synthesis.  The motivation is to improve the accuracy with which the resulting generated images reflect the shape of a viewed object viewed from a particular angle, and to improve the generality of camera placement for the new image. Specifically the authors seek to encourage shape and texture consistency between multiple synthetic views of a single generated object, rather than have the model generate different shapes for each new view. They also aim to have the model render images that accurately reflect the impact of a given change in camera pose. The proposed model extends the Zero123 model by finetuning with altered loss functions that reflect their goals. They particularly introduce a closed-loop cost which compares rendered new views against ground truth.

**Strengths:**

The experiments show that the method improves the performance measures chosen, and that the proposed method generates images that depict more consistent geometry than do the images generated by Zero123.

The paper offers the following insights which are of interest:
- path features are more informative than class features in training new view synthesis models
- applying an MSE-based loss in the latent space of an autencoder leads to increasing vulnerability to training collapse
- Section 3.1 provides a good explanation of current NVS methods

**Weaknesses:**

The primary contribution of the paper is a method for finetuning Zero123. The fact that the proposed method applies only to Zero123 means it is of limited significance, or interest to the rest of the field.

The change to Zero123 is cosmetic rather than fundamental, and is not transferable to other methods.

The presentation of the paper is good and bad.  It is well laid out, and effort has definitely been applied to survey the relevant literature. The English and the mathematics are very difficult to read, however, to the point where the Introduction is hard to parse.  This is particularly true of the literature review in the Introduction which lists many papers but leaves the reader more confused than illuminated. Effort has been made with the maths, but there is no consistency.

Some indicative examples of the problems with the presentation:
- There are at least 3 problems with the first sentence of the body of the paper: "Recent advancements in novel view synthesis (NVS) have sparked considerable excitepment on 3D generation"
- "Although the modified task settings, like generating fixed-view multiple views"
- The first three mathematical quantities introduced are \bf{X}, \bf{\Delta R T}, and \bd{X}_{tg}. These are latter explained to be an image, a tuple of matrices, and an image, respectively, despite the fact that they have the same notation. Using a symbol to mean one thing, then the same symbol with a subscript to mean something different is challenging.
- Line 211 introduces a decoder 'g()' as being parameterised by \eta, despite there being no \eta shown.
- Equation 3 minimizes over \eta^* despite the fact that this variable is not a parameter to the expression to be minimized
- g() has variable numbers of parameters, and none of them are \eta^*

**Questions:**

I don't have any questions

---

### Meta-Review · Area_Chair_DKq1 · 2024-12-21

**Metareview:**

The paper introduces Ctrl123, which aims at improving multi-view consistency in novel view synthesis by aligning generated views with ground truth in the CLIP patch feature space. It extends Zero123 by fine-tuning with modified loss functions, including a closed-loop transcription cost. This approach enhances multi-view consistency while supporting arbitrary camera placements.

The common strengths identified: (1) its innovative use of CLIP-based patch features and closed-loop transcription to enhance multi-view consistency; (2) its flexibility to generate arbitrary views while improving NVS quality; and (3) its extensive experimental validation and new metrics like Angle Accuracy and Mask IoU.

The major weaknesses include: (1) the method is a minor tweak to Zero123, lacking significant novelty or broader applicability; (2) high computational costs due to unrolling in training; (3) unclear presentation, with missing supplementary materials like videos to demonstrate multi-view consistency; and (4) no comparative analysis with Zero123++ or inclusion of key metrics like LPIPS.

The overall ratings are consistently below the broardline and the rebuttal is missing, thus, the ACs agreed on rejection.

**Additional Comments On Reviewer Discussion:**

No rebuttal was found.

---

### Decision · Program_Chairs · 2025-01-22

Reject